# Neural Regulations in Tooth Development and Tooth–Periodontium Complex Homeostasis: A Literature Review

**DOI:** 10.3390/ijms232214150

**Published:** 2022-11-16

**Authors:** Yihong Duan, Yongfeng Liang, Fangyi Yang, Yuanyuan Ma

**Affiliations:** 1Hospital of Stomatology, Guangdong Provincial Key Laboratory of Stomatology, Sun Yat-sen University, Guangzhou 510055, China; 2Guanghua School of Stomatology, Sun Yat-sen University, Guangzhou 510055, China

**Keywords:** tooth, neural regulation, development, homeostasis, periodontium

## Abstract

The tooth–periodontium complex and its nerves have active reciprocal regulation during development and homeostasis. These effects are predominantly mediated by a range of molecules secreted from either the nervous system or the tooth–periodontium complex. Different strategies mimicking tooth development or physiological reparation have been applied to tooth regeneration studies, where the application of these nerve- or tooth-derived molecules has been proven effective. However, to date, basic studies in this field leave many vacancies to be filled. This literature review summarizes the recent advances in the basic studies on neural responses and regulation during tooth–periodontium development and homeostasis and points out some research gaps to instruct future studies. Deepening our understanding of the underlying mechanisms of tooth development and diseases will provide more clues for tooth regeneration.

## 1. Introduction

Innervation regulates craniofacial development and homeostasis while being reciprocally affected by target tissues. For example, the absence of innervation causes target organ deformation, as in individuals with Moebius syndrome, a disease caused by the absence or underdevelopment of VI and VII cranial nerves, a series of craniofacial deformations is present [1]. Rabbit submandibular glands undergo atrophy upon denervation, which can be reversed by redistribution of autonomic nerves [2].

The tooth–periodontium complex, including teeth, gingiva, periodontal ligament (PDL), and alveolar bone, is one of the densely innervated cranial structures containing trigeminal sensory nerves and autonomic nerves. An array of evidence has demonstrated the mutual regulation of tooth and dental nerves in development [3]. The mutual regulation also exists in tooth and periodontal homeostasis. For instance, innervation orchestrates inflammation [4], promotes periodontal healing [5], and regulates the physiological reparation and regeneration of dental tissue [6]. These interactions in development and diseases are mediated by a variety of nerve- or tooth-derived molecules and will probably inspire advances in tooth regeneration.

Currently, researchers have employed various strategies for tooth regeneration using fetal tissue [7,8] and tooth germ cells [9] or using adult mesenchymal stem cells (MSCs) [10]. They imitate tooth development and physiological reparation, respectively. However, the innervation of bio-engineered teeth does not fully replicate the highly asymmetric and organized pattern of the natural tooth–periodontium complex [7]. Better restoring tooth innervation requires deeper understanding of the mutual regulation between tooth and dental nerves in development and homeostasis. Understanding this will also benefit dental tissue engineering, as some neurogenic molecules are effective in promoting dental tissue formation [11].

In this review, we provide a panoramic view of recent findings on neural responses and regulation in tooth development, growth, and homeostasis and some related applications in tooth regeneration. We also try to point out research gaps for future studies to fill. Our review will provide more clues for improving tooth regeneration.

## 2. Anatomical and Embryological Basis

A tooth can be divided into two parts: the crown visible in the mouth after eruption and the root anchored in alveolar bone by the periodontal ligament. The dental pulp, within the tooth center, is extensively filled with nerves composed of sensory and sympathetic fibers, which arise from the trigeminal ganglion (TG) and the superior cervical ganglion (SCG) respectively (Table 1). Whether parasympathetic nerves innervate the dental pulp is still a subject of debate, but parasympathetic neuropeptides are observed in the dental pulp [4]. The axons mainly accumulate in the coronal pulp to form the subodontoblastic Raschkow plexus, implicating possible crosstalk between odontoblasts and pulpal nerves [4]. The periodontal ligament is densely innervated by mainly sensory fibers and a few autonomic fibers (Table 2). The sensory endings include various proprioceptors, such as Ruffini ending, differing from the sensory innervation in the pulp. A brief summary of pulpal and periodontal nerves is presented in the tables below.

In recent years, extensive studies have suggested the significance of local sensory and sympathetic nerves in maintaining dental and periodontal health and promoting the recovery of local diseases. This not only is because of their anatomical localization but also is demonstrated through inferior alveolar nerve resection (IANr), which not only reduces incisor dentin/enamel formation and injury repair, but also suppresses alveolar bone remodeling and periodontal defect repair [12,13,14]. In addition, sympathetic innervation also appears to take a regulatory role in osteoclast infiltration in rat periapical lesions [15], and alveolar bone remodeling in oral challenge with *P. gingivalis* [16]. Reversely, carotid sinus nerve stimulation plays a protective role in rat periodontitis, attenuating alveolar bone loss and inflammation [17]. Altogether, these studies agree that normal sensory and sympathetic innervation are critical for tooth–periodontium complex.

The tooth is also capable of influencing nerves, which will be discussed in detail in the following section. The foundation of this mutual regulation between nerves and the tooth–periodontium complex probably lies in their embryological origin. The dental mesenchyme origins from the cranial neural crest (CNC) cells and is developmentally associated with the neural system. One study has revealed that the mesenchymal stem cells of dental pulp (DPSCs) can still be differentiated into functional dopaminergic neurons by midbrain cues in vitro [18]. This remaining responsiveness indicates dental mesenchymal cells have preserved some hereditary similarity with neural cells. More convincingly, RNA-seq analysis has revealed neural/glial cells have a closer origin relation with skeletal cells than pigment cells on the genetic level [19]. This developmental homology suggests there may still be reciprocal regulation between the tooth and the nervous system.

## 3. Tooth Influences Neurophysiology during Development Process

### 3.1. Tooth Innervation Is Spatiotemporally Regulated

The tooth sequentially develops as the result of epithelial–mesenchymal interactions. During the embryonic (E) stage for crown formation, the tooth experiences bud, cap, and bell stages, followed by root formation including crown–root transition, root elongation, eruption, and full development after birth (Figure 1). Tooth innervation happens concomitantly with tooth formation, and appears to strongly correlate with tooth developmental stage.

At E11.5, mouse molar tooth formation is initiated as a local thickening of dental epithelium, while the pioneering trigeminal fibers first arrive at the maxillary oral epithelium at E10 [20]. However, the trigeminal fibers do not infringe on the presumptive dental mesenchyme or developing tooth germ but only navigate their periphery from initiation to the cap stage and grow into the dental follicle (DF) at the bell stage [20]. At the cap stage, the sensory fibers form a plexus under the dental papilla, which later gives rise to the fine DF branches, but will not enter the pulp until a thin layer of enamel has been deposited at postnatal (PN) day 4. Intriguingly, the trigeminal fibers enter the pulp through presumptive root apices in a multi-rooted tooth, in the absence of any physical barrier such as the pulp floor [20], implicating a predetermined mesenchymal route for nerve ingrowth. The sympathetic fibers, which previously only exist around blood vessels outside the dental pulp, enter the dental pulp following sensory fibers on PN9, after root formation has begun [21]. This indicates that the ingrowth of dental nerves follows developmentally regulated timing, order, and routes.

Evidence shows that the developing dental tissues can regulate their own innervation by either attracting or repelling neurites (Figure 1). For example, semaphorin 3a (Sema3a) is a nerve repellent expressed in tooth germs. Its distribution in tissue shifts as tooth development proceeds, thus controlling the timing and route of tooth innervation [20]. On the other hand, auto-transplanted tooth germs can induce their own innervation [22], and adult denervated teeth can be automatically reinnervated [23,24]. In line with these findings, an in vitro co-culture study reveals tooth germ mesenchyme differentially attracts or repels neurites from TG explants depending on the developmental stage [25], meaning both the attracting and repelling are developmentally regulated.

### 3.2. The Molecular Guidance Cues for Tooth–Periodontium Innervation

Many molecules, mainly neurotrophins and semaphorins, expressed in and around the developing tooth germ modulate tooth innervation and other aspects of tooth development, such as cytodifferentiation. Many of them are also implied in periodontal innervation. Studies of tooth regeneration do not monitor the expression of these molecules [7,9,10], but their proper spatiotemporal expression is vital for the normal patterning of tooth nerves. Proper application of these molecules may benefit tooth nerve regeneration.

#### 3.2.1. Neurotrophins

Neurotrophins are identified as a family of growth factors important for neuronal survival, development, and function, as well as for the immune and reproductive systems. NGF as a prototypic neurotrophin is a significant target-derived promotive factor for peripheral sensory and sympathetic innervation. NGF immunoreactivity is mainly located in the dental mesenchyme in the bud stage, while in the cap and bell stages, it is mainly expressed in the dental epithelium and adjacent odontoblast layer. Its gene expression pattern correlates with pulpal neurite growth in postnatal mouse molars, suggesting that NGF is involved in the local sprouting, guidance, and arrangement of trigeminal axons in developing teeth [26]. This can explain why neonatal exposure to anti-NGF reduces the number of trigeminal neurons projecting to rat molar dental pulp [27].

NGF receptors TrkA and p75NTR (Figure 1) are concentrated in nerve fibers approaching the tooth germ and mediate innervation throughout the whole development process [28]. It is reported that TrkA knockout mice lack both sympathetic and sensory pupal nerves [29] and the deletion of p75NTR in mice causes weaker osteogenic ability [30]. Furthermore, the p75NTR gene positively correlates with mineralization-related genes in mouse ectomesenchymal stem cells [31], possibly involved in the melanoma-associated antigen-D1 (Mage-D1) and NF-κB pathway [32,33]. Therefore, the dental NGF receptors are important not only for innervation but also for tooth morphogenesis and hard tissue formation.

Other neurotrophic factors, including brain-derived neurotrophic factor (BDNF), neurotrophin-3 (NT-3), and glial-derived neurotrophic factor (GDNF) are also implicated in the innervating of dental nerves (Figure 1). Ablation of GDNF/Ret signaling before pulpal innervation significantly reduces pulpal neurites in Ret knockout mice, as well in mice treated with Ret inhibitor [34]. Interestingly, in an adult tooth, Ret^+^ fibers only make up a portion of pulpal afferents and are predominantly expressed in a mutually exclusive manner with TrkA in trigeminal sensory neurons, although all the small, medium, or large diameter trigeminal sensory neurons may contain them both concomitantly [34]. Even so, GDNF and NGF work in synergy to promote the innervation of all types of neurons classified by size. However, their individual target neurons have little overlapping, meaning they cannot be regarded as functionally equivalent agents. A similar phenomenon might also exist with other neurotrophins mentioned above. Recently, increasing studies are pointing to the morphological and even functional heterogeneity among dental pulp afferents [35,36,37,38]. Therefore, neurotrophins may have a predilection for specific subpopulations of pulp afferents with a mutually exclusive expression.

Neurotrophins are also involved in periodontal innervation. Periodontal mechanoreceptive Ruffini ending shows immunoreactivity for TrkB, an affinity receptor for BDNF and neurotrophin-4/5. BDNF knockdown reduces periodontal nerve density and causes Ruffini ending deformation in mice [39] more severely than neurotrophin-4/5 depletion does at the early stage [40]. The cellular distribution and expression patterns of GDNF and its receptors in the periodontium coincide with the development of Ruffini endings. Ablation of GDNF/Ret signaling also results in abnormal periodontal fibers [34]. This indicates that various neurotrophins regulate periodontal innervation.

#### 3.2.2. Semaphorins

The semaphorin family is a class of secreted, membrane-bound proteins that regulate many developmental processes [41]. Most semaphorins are chemorepellent against neurites, but some may promote axonal growth, such as semaphorin7A, which is expressed in odontoblasts and promotes dentin–pulp complex terminal innervation.

Sema3a is the most studied in dental development studies. Sema3a expression is absent from the distribution domains of dental nerves in embryonic mouse tooth germ, while the expression of its receptor, neuropilin-1 (Npn1), is restricted to dental axons. In *Sema3a^−/−^* and *Npn1^−/−^* mice embryos, nerve fibers appear prematurely in the presumptive dental mesenchyme and later ectopically in the dental papilla mesenchyme [42,43]. In postnatal mouse molar, Sema3a is observed in the future pulp floor area, but absent from the future apical foramen, which illustrates its control of normal pulpal nerve pathways through the apical foramen [20]. Interestingly, the expression pattern of multiple neurotrophins is not altered in *Sema3a^−/−^* mice, indicating the attracting and repelling regulations in tooth innervation are exclusive but coordinated [20,42,43].

Sema3a also impacts periodontal nerve distribution and structure. In *Sema3a^−/−^* mouse incisors, the number of nerves and arborizations abnormally increases in the developing dental follicle target field and periodontium, and there are abundant nerve fibers in the labial periodontium [42]. Unlike in the periodontium, whether Sema3a affects pulpal nerve structure is contingent. In Sema3a null mice, pulpal nerve thickness and fasciculation are notably reduced in molars but have no apparent changes in incisors [42,43]. This is probably due to the subtle differences between incisor and molar innervation, which is also the explanation for why Sema3a-knockout-induced premature innervation happens slightly later in incisors than in molars [42].

Notably, Sema3a modulation is discrepant between teeth and other hard tissue. Its knockout does not trigger dental hard tissue defects but causes skeletal deformation and osteopenia [44]. Speculatively, this is probably because other semaphorins that possess similar functions, mainly class 3, 4, 5, and 6 molecules [25], also potentially regulate tooth development and may compensate for the lost non-neuronal function of Sema3a in developing teeth.

The upstream signaling of Sema3a is seldom studied. It appears that Wnt4 and Tgf-β1 in the dental epithelium induce early Sema3a expression in the dental mesenchyme on E10-E11 via epithelial–mesenchymal interactions (Figure 1). In addition, *Fgfr2b^−/−^* molar germ is devoid of Tgf-β1, which accounts for the abnormal Sema3a expression from the E13.5 late bud stage onwards [45]. The canonical Wnt signaling pathway is also altered in *Fgfr2b^−/−^* teeth, but not via Wnt4. Evidence shows that Wnt4 stimulates jaw mesenchymal Msx1 expression, which is indispensable for tooth morphogenesis beyond the bud stage, whereas Tgf-β1 promotes dental mesenchymal cell proliferation. Thereby Wnt, Tgf-β, and Fgf signaling conjugate tooth innervation with tooth formation [20].

Conclusively, Sema3a is a potent repellent for pulpal and periodontal innervation. Local injection of a Sema3a inhibitor alone is sufficient to promote innervation of bio-engineered teeth [46], revealing its potency as a regulatory target for bio-engineered teeth.

### 3.3. Related Application in Tooth Regeneration

The molecules controlling tooth innervation may be applied for achieving better regeneration of the tooth neural network. In a study of preserving immature teeth with non-vital pulp, the application of amelogenin after root canal treatment successfully induces pulp regeneration with a dense neural network [47]. Amelogenin is a vital component of the enamel matrix. This confirms molecules other than neurotrophins and semaphorins may also regulate tooth innervation and again demonstrates that a developing tooth controls its own innervation.

Similarly, neurotrophins and semaphorins may also be applied in tooth and periodontium regeneration. However, this study, like many other regeneration studies, did not compare the distribution, pattern, and function of the regenerated pulp neural network with those of a natural one, meaning the regeneration may not be fully successful. Therefore, if neurotrophins and semaphorins are really to be applied, their physiological expression pattern during tooth development should be carefully scrutinized to instruct proper application so that the tooth neural network can best mimic a natural one.

## 4. The Nervous System Regulates Tooth Development

### 4.1. Neural Regulations in Tooth Development at Pre-Eruptive Stage

Controversy exists with regard to the participation of nerves in tooth initiation. In polyphyodont fish, denervation completely abolished tooth germ formation. However, studies on mice suggest the opposite. Mouse diastema is the edentulous region between incisors and molars which contains three unerupted tooth primordia that develop until the bud stage. It is devoid of peripheral nerves, indicating innervation is not required for mouse tooth initiation [48]. Consistently, mouse embryonic mandible explants with or without TG attached produce similar tooth formation incidence [49]. However, these conclusions are flawed in that they cannot factor out the pioneer trigeminal fibers that arrive near the oral epithelium prior to dental placode formation [50]. These pioneer fibers might prime the oral epithelium to form a dental placode (Figure 1). In fact, a recent study detected calcitonin gene-related peptide (CGRP), a common sensory nerve marker, from E14.5 to E17.5 with epithelium predilection in mice [51]. In addition, substance P (SP), a well-known member of the tachykinin family released by sensory neurons, is transiently expressed in developing mouse tooth germ. Its receptor antagonist blocks the development of ex vivo mouse tooth germs at E14, which is reversed by exogenous SP [52]. These pieces of evidence endorse the speculation that some early-arriving sensory fibers interact with the oral epithelium to initiate tooth development. Above all, controversy persists on the role of innervation in tooth initiation and early development before mineralization. In view of the pioneering trigeminal fibers to the oral epithelium, it would be desirable to use animal models whose dental innervation is completely ablated with genetic tools to obtain more compelling evidence.

In the mineralizing phase of tooth development, innervation clearly regulates the normal structure and morphology of the tooth. In cap and bell stage mouse molars, CGRP is mainly detected around the dental epithelium–mesenchyme interface, and its expression suddenly increases during E17.5 to E18.5, when odontoblasts and ameloblasts differentiate from the dental papilla and inner enamel epithelium, respectively [51]. This coincidence indicates a role of sensory fibers in tooth development, but how the nerves elicit normal dentinogenesis and amelogenesis remains to be explored.

Some studies have highlighted specific neurogenic factors in the regulation of tooth development. Nerve fibers containing PACAP, a pleiotropic neuropeptide, have been observed in the odontoblastic and subodontoblastic layers of the dental pulp [53] (Figure 1). PACAP-deficient mice exhibit aberrant enamel or dentine structure, accompanied by smaller incisors and thinner molar dentin [54]. This is most likely due to the regulation of PACAP on Sonic hedgehog (Shh) and Notch signaling which are critical in tooth morphogenesis [55,56].

In addition, neurohormones released in the central nervous system (CNS) also regulate tooth development. Melatonin can enhance odontogenic differentiation of dental papilla cells in the late bell stage [57] by upregulating RORα and altering mitochondrial function and biogenesis in these cells [58,59]. Melatonin is also reported to promote the normal survival and mineralization of ameloblasts in postnatal mouse molar germs [60] by activating the MAPK signaling via repressing β-arrestin [61], and the Wnt pathway may also be involved [62]. Moreover, the precursor of melatonin, serotonin, stimulates the development of tooth germs in an embryonic mouse mandibular explant culture beyond the bud stage, meaning it may promote the differentiation of tooth germ cells [63]. Further analysis reveals this may be achieved by direct activation of serotonin receptors [64]. In addition, serotonin 2B receptor knockdown leads to abnormal enamel volume and structure in mouse molars [65]. Serotonin regulates periodontal development too. The intake of a serotonin re-uptake inhibitor, fluoxetine, decreases periodontium-forming cells [66].

Together, these studies suggest the importance of innervation and neuron-derived factors in regulating normal tooth morphogenesis. These factors may prove helpful in regenerating teeth with tooth germ cells or fetal tissue with respect to their developmental roles.

### 4.2. Neural Regulations in Tooth Eruption

Many hypotheses have emerged trying to elucidate the underlying mechanism for tooth eruption, but none have gained entire recognition. However, it is widely accepted tooth eruption depends on the development and remodeling of the surrounding jaw bone and the extent of tooth root development [67]. Therefore, neuroregulation probably regulates eruption by targeting these physiological processes (Figure 1).

*Alveolar bone remodeling*: Both sensory and autonomic nerves can actively regulate bone remodeling. During experimental tooth movement, nerve bundles are found to co-localize with osteoclasts and Howship’s resorption lacunae [68], providing ponderable evidence of neural participation in alveolar bone resorption. In light of this, alveolar nerves might also participate in tooth eruption. Although the researchers did not further specify the biomolecular event in this study, intrabone nerves regulate bone remodeling mainly through secreting various molecules targeting bone cells [69]. For instance, the sympathetic neurotransmitter norepinephrine overall suppresses bone formation and promotes bone resorption [69].

*Dental follicle (DF)*: The DF plays a critical role in tooth eruption by regulating jaw bone remodeling and tooth root growth. Eruption totally fails after surgically removing DF, while the removal of the developing tooth crown or root did not disrupt normal eruption [70]. However, a direct investigation into the neural regulation of DF function is still absent to date. Nevertheless, it seems dental innervation can regulate DF function indirectly via Hertwig’s epithelial root sheath (HERS), an important guide for root growth. Growing HERS expresses a receptor for vasoactive intestinal peptide (VIP), which enhances HERS elongation in ex vivo tooth germs by promoting cell proliferation [71]. In turn, HERS cells can stimulate the differentiation of cementoblasts or osteoblasts and the mineralization of DF cells via the Wnt signaling pathway [72].

*Local blood flow*: The vasculature within the alveolar socket is also thought to affect tooth eruption. The local injection of vasoactive drugs alters socket blood flow as well as tooth eruption rate in rats dose-dependently [73]. Systemic volume expansion by injection of Ringer’s solution also increases socket blood flow and tooth eruption rate by decreasing systemic arterial pressure, not by local nutrition [74]. Both sensory and autonomic innervations in the periodontium regulate local blood flow, thereby regulating tooth eruption.

Overall, although neural participation in tooth eruption is abundantly implicated, a concrete mechanism is still waiting to be unveiled, and further research in this direction will help decipher the mystery of the general tooth eruption mechanism as well. For rodent incisors, tooth eruption is continuous throughout the whole life. We will discuss stem cells and neural regulation in incisor homeostasis and injury repair in detail in Section 7.

### 4.3. Related Application in Tooth Regeneration

Some of the neurogenic molecules implicated in tooth development are applicable to tooth regeneration, imitating their developmental roles. For example, serotonin is used to treat murine oral keratinocytes (MOKs), which also arise from embryonic mouse oral epithelium as the enamel organ does. MOKs growing in the presence of serotonin after reprogramming with Tgf-β1 successfully produced amelogenin, and in a 3D culture, they formed enamel-producing organoids. After densification with pressure and vacuum, the enamel formed by MOKs possesses a similar structure and mechanical properties to natural enamel, suggesting the potential application of serotonin in enamel regeneration via tissue engineering methods [11]. Other molecules discussed here may also help regenerate dental tissues, but their efficacy and application methods need to be explored.

## 5. Dental Disease Causes Neurophysiological Changes

### 5.1. Morphological Changes in Local Dental Nerves

Myelinated nerves in the apical pulp of human teeth with irreversible pulpitis undergo various kinds of degenerative changes. The sprouting and degeneration of nerves are initial events occurring in carious or pulpitis teeth, and the exact mechanisms are still under investigation. However, the nerves do not just passively wait to be damaged by inflammation. In carious teeth, overexpression of growth-associated protein 43 (GAP43), a protein that promotes extensive sprouting of nerve endings into the reparative dentin matrix, is observed [75]. As shown in Figure 2, caries-induced pulp inflammation activates the complement system resulting in the production of C5a [76]. In addition, all pulp fibroblasts beneath the carious site express the C5a receptor (C5aR), which activates the production of BDNF upon binding to C5a. In turn, BDNF guides the outgrowing neurites towards the carious site [76]. However, C5a/C5aR binding seems to downregulate pulp fibroblasts’ production of NGF, another neurite outgrowth inducer that mainly promotes neurite length rather than quantity. Nevertheless, pulp fibroblasts’ C5aR activation is still essential for augmented neurite outgrowth in carious conditions [77]. Contrastingly, DPSCs are shown to overexpress both BDNF and NGF in response to C5a [78]. The NGF overexpression is shown to increase nerve density, which will subside once healing occurs but continue if inflammation persists, suggesting neural regulation in dental disease progression [79]. NGF is also shown to elicit the neuronal differentiation of DPSCs, which might also contribute to the regeneration of injured nerve fibers [80]. These simultaneous actions of immune, pulp, and neural cells suggest a coordinated neuroimmune response in host defense and healing. Accordingly, molecules mentioned here may be applied to better regenerate pulp innervation.

### 5.2. Molecular Changes in Neural Cells

Molecular changes in dental innervation are also observed in various aspects. Voltage-gated sodium channels (NaVs) are transmembrane ion channels on sensory neurons whose expression is altered in caries and pulpitis. In particular, NaV1.7, NaV1.8, and NaV1.9 are associated with aggravated pain or inflammation [81,82]. In contrast, Kv1.4, a voltage-gated potassium channel suppressing the generation of action potential, is downregulated on sensory neurons during pulpitis [83]. In addition, transient receptor potential channels (TRPs), nonselective cation channels on sensory nerves, activated by inflammatory agents or bacterial products are also involved in the generation of pain in pulpitis, as their activation causes the release of CGRP, a pain mediator [84,85]. Interestingly, in spite of local neural changes, distant neurons in the ipsilateral trigeminal ganglion of inflamed pulp also display pathological changes [86].

Neurosecretion also changes in pathological conditions. A variety of neuropeptides, including neurokinin A, substance P (SP), NPY, and CGRP, are shown to be upregulated in the pulp in conditions of caries, pulpitis, or occlusal trauma [87,88,89]. These neuropeptides still have other physiological functions contributing to the regulation of tooth and periodontal homeostasis, which reflect the mutual regulation between nerves and target oral tissues (Figure 2) and will be described in detail in Section 6.

### 5.3. Related Application in Tooth Regeneration

Molecules inducing neurite outgrowth in dental diseases might be applicable to tooth regeneration for better re-innervation outcomes. One study used rat bone marrow mesenchymal stem cells (BMSCs) incorporated in a poly L-lactic acid scaffold to regenerate dental pulp in pulpotomized rats. The bio-engineered pulp was successfully re-innervated with increasing expression of NGF and GAP43 in the tooth, but at the end of the study, the nerve fiber density was still lower than that in a natural tooth [90]. This implies molecules promoting nerve growth can probably be mixed into the scaffold to achieve improved re-innervation outcomes.

## 6. The Nervous System Influences the Pathology of Dental Diseases

The neural changes have important effects on the progression of pathological conditions by regulating the activity of target cells, which is mainly mediated by neurogenic factors including SP, CGRP, NPY, VIP, PACAP, and acetylcholine (Ach) in the context of dental diseases. Multiple neuropeptides often work together, either synergistically or antagonistically, and interact with each other to regulate tooth and periodontal homeostasis (Figure 2). On the other hand, the states of tooth–periodontium complex also affect the production of peripheral neurotransmitters, and the reciprocal action between them is complicated. In addition, in recent years, studies have also revealed the emerging role of some neurohormones released in the central nervous system in tooth and periodontal homeostasis. All these studies might provide inspiration for tooth–periodontium regeneration.

### 6.1. SP

SP is an 11 amino acid neuropeptide, a well-known member of the tachykinin family, produced by a variety of cells including sensory neurons, macrophages, dendritic cells, eosinophils, and lymphocytes [91]. The production of SP is upregulated in many pathological conditions and dental treatments. There is evidence in orthodontic movement showing an increase in SP resulting from the NGF/TrkA-PDL-associated sensory neuron body [88,92].

Dental pulp fibroblasts express both SP and its receptor, neurokinin-1 receptor (NK-1), and SP expression is significantly upregulated in carious teeth and irreversible pulpitis [87,93], suggesting its regulation in pulpal neurogenic inflammation, and it acts at least in part in an autocrine fashion [93]. Mechanistically, SP induces IL-8 and MIP-1 production in dental pulp [94,95] and TNF-a, IL-1b, IL-2, and IL-6 production in rat cutaneous wounds [95] and therefore is involved in the orchestration of pulpal inflammatory response. SP/NK-1 signaling is also significant in pulpal pain generation [96], as SP is a pain mediator, and NK-1 expression in inflamed dental pulp is tightly correlated with the degree of pain. Furthermore, SP is a vasodilator in pulp, activating NO synthase for NO production from vascular endothelial cells [97]. It also increases blood flow via the release of histamine [98]. This partially accounts for the hyperemia observed in inflamed pulp.

In the periodontium, SP is conducive to gingival inflammation and periodontal bone destruction. SP in gingival crevicular fluid (GCF) is reduced as the periodontal inflammation is relieved [99], meaning SP may be used as a predictor for disease progression. SP also promotes osteoclast differentiation by upregulating the RANKL/OPG ratio in gingival fibroblasts, and it upregulates hypoxia-inducible factor-1 alpha (HIF-1α) in macrophages and gingival fibroblasts, leading to inflammation and angiogenesis [100]. These findings together reveal SP’s action in periodontitis.

SP seems to promote pulpal and periodontal inflammation and related tissue destruction. Interception of local SP signals with a drug will probably benefit the outcome of tooth and periodontal regeneration in inflammatory diseases.

### 6.2. VIP

VIP is a potent anti-inflammatory parasympathetic neuropeptide distributed throughout the central and peripheral nervous systems (PNSs). The biological actions of VIP are mediated by two high-affinity receptor subtypes, VPAC1 and VPAC2, on target cell membranes. VIP and VPAC1 expressions are elevated in moderately carious teeth to protect the pulp [101]. In periapical granulomas, VIP is a therapeutic target whose expression is associated positively with immunoregulatory factors and inversely with proinflammatory mediators [102]. The similar functions of VIP are also shown to alleviate inflammation and bone loss in periodontitis through regulating immune responses via monocyte differentiation, Treg migration, or TNF-α production and in osteoclastogenic response via the RANKL/OPG ratio [102,103,104,105]. These data suggest VIP may help eliminate local inflammation to create better conditions for tooth and periodontium regeneration.

### 6.3. ACh and Cholinergic System

The cholinergic system, including ACh, choline acetyltransferase (ChAT), acetylcholinesterase (AChE), and nicotinic or muscarinic acetylcholine receptors (AChRs), is involved in periodontal disease. Ach, an important neurotransmitter, is mainly expressed by the vagus nerve. Although little evidence shows that the vagus nerve directly affects the periodontium, the components of a functional non-neuronal cholinergic system have been found in oral mucosa [106]. ACh can modulate innate immune responses within the oral mucosa in both an autocrine and a paracrine manner and subsequently participate in the pathological process of periodontal disease [107]. A clinical study showed that the Ach level increases significantly in saliva and GCF with higher levels of IL-17A and IL-17F, which are correlated positively with the severity of periodontal diseases [108], meaning the non-neuronal cholinergic system may influence the etiopathogenesis of periodontal disease.

Nicotine also plays an important role in periodontal diseases. The use of tobacco products is closely associated with an increased incidence of periodontal disease and unsatisfactory periodontal treatment. It is well established that a variety of alpha 7 nicotinic acetylcholine receptors (nAChRs) are expressed in human periodontal ligament cells (hPDLCs) and rat periodontal tissues. Nicotine activation of nAChRs induces IL-8 release as the gingival epithelial innate immune response [109]. Recently, nicotine/α7 nAChR signaling has been shown to regulate autophagy of hPDLCs, and subsequently promote the release of inflammatory factors including IL-1β and IL-8 [110]. These findings, therefore, suggest a direct proinflammatory action of nicotine/nAChR on the pathogenesis of periodontal disease. Intercepting nicotine/nAChR signals with topical drug delivery may improve the periodontal regeneration outcome of conventional periodontal treatment.

### 6.4. PACAP

PACAP is a neuropeptide that is widely distributed in and protects not only the nervous system but also peripheral organs [111]. PACAP and its specific receptor PAC1 regulate periodontal bone remodeling after tooth luxation, where PACAP-positive nerve fibers evidently increase and are located in close proximity with PAC1-positive osteoclasts and later-appearing osteoblasts [5]. In addition, PACAP is also shown to have an immunomodulatory effect in periodontal wound healing after tooth extraction, which is to elicit an M2 phenotype of macrophages. However, endogenous PACAP production is mitigated by exogenous PACAP supplementation, and the bone healing does not correspondingly improve in rats. This is probably because PACAP as an additional healing signal will not improve already optimal healing in homeostatic conditions [112]. Above all, despite in vitro studies’ emphasis on the regulatory role of PACAP, its therapeutic potential in regeneration is still questionable and should be subjected to further scrutiny.

### 6.5. NPY

NPY is a sympathetic neurotransmitter with important functions in various physiological processes and diseases. Its receptors include neuropeptide Y Y1-6 receptors (NPY Y1-6R) [113]. In human teeth and periodontium, NPY Y1R is the most studied.

NPY and NPY Y1R, distributed around small blood vessels in dental pulp, regulate caries-induced pulpal inflammation by modulating blood flow and inflammatory agent production. In pulpitis, NPY Y1R is also found in infiltrating inflammatory cells, nerve fibers, and dental pulp fibroblasts, implicating its regulation of neurogenic inflammation [114]. NPY Y1R on dental pulp fibroblasts is upregulated in response to inflammatory cytokines (IL-1β, Tgf-β1) and neuropeptides (SP, NPY), thus modulating the inflammation in turn [115]. On trigeminal ganglion (TG) neurons, NPY Y1R co-localizes with transient receptor potential channel vallinoid1 (TRPV1), which controls the release of CGRP. NPY binding to it inhibits the release of CGRP in the dental pulp to modulate neurogenic inflammation [116].

NPY also plays an important role in periodontal homeostasis. Inflamed human periodontal ligament stem cells become more sensitive to NPY, which enhances their osteogenic capacity in vitro [117]. Moreover, a genome-wide association study found suggestive evidence of the association of NPY loci with severe chronic periodontitis [118]. This is consistent with another genome-wide study, which also shows that single-nucleotide polymorphisms (SNPs) upstream NPY have strong interaction with sex, explaining the risk predilection of severe periodontitis in men [119]. These findings recently have been verified in a Brazilian population [120]. Therefore, NPY is significant in both the regulation and the initiation of periodontitis. However, direct experimental evidence will be required to discuss whether NPY supplementation helps periodontal regeneration, as current basic studies only revealed the correlation but not the nature of this correlation.

### 6.6. CGRP

CGRP is a neuropeptide associated with neurogenic inflammation and painful orofacial conditions. During pulpitis, the expression levels of CGRP and CGRP receptors are upregulated, relating to dental pain and other histopathological phenomena [121]. CGRP receptors are expressed on a large portion of M2 macrophages in irreversible pulpitis or healthy pulp, indicating the immunomodulatory function of CGRP [122]. Consistently, LPS from *Porphyromonas gingivalis* (Pg), a pathogen commonly associated with symptomatic pulpitis, can directly excite trigeminal ganglion neurons via transient receptor potentials (TRPs) sensitized by toll-like receptor 4 (TLR4), resulting in CGRP production [123]. Axonal CGRP is commonly elevated in peripheral nerve injury and is involved in the regeneration of local nerve fibers by promoting Schwann cell proliferation, meaning the increased CGRP production might also promote pulpal nerve regeneration in pulpitis, where pulpal nerves are damaged by inflammation and bacterial products [124]. In addition, in vitro studies reveal it antagonistically but coordinately regulates the formation of reparative dentin with Shh, as they are secreted from sensory afferents in combination upon stimuli [125]. However, CGRP regulation is variable across individuals, explaining the variable levels of pain and disease severity among patients with caries or pulpitis [89]. In apical periodontitis, capsaicin-sensitive nerve fibers exert a protective effect against the progression of periodontal lesions, possibly relating to the release of CGRP and other neuropeptides [126]. Overall, CGRP holds the potential for tooth regeneration, but its efficacy may depend on individuals.

CGRP protects against the progression of periodontitis. Most neurons in TG that innervate the rat gingival mucosa contain CGRP and SP. CGRP inhibits the apoptosis of osteoblasts induced by LPS [127] and inhibits osteoclastogenesis in vitro, both contributing to the alleviation of bone loss [128]. However, CGRP content is significantly reduced in GCF in the inflamed periodontium, where the defense and regenerative capacity is debilitated [129]. Supplementing CGRP may help recover this regenerative capacity. CGRP regulation is also implicated in other periodontal conditions, such as in occlusal trauma combined with moderate orthodontic forces [88], and the mutual interaction of CGRP and Shh promotes socket healing after tooth extraction [130]. In summary, CGRP has a substantial influence on various periodontal conditions that may be helpful in regenerative therapy.

### 6.7. CNS-Released Neurohormones

Aside from the above-listed neuropeptides from peripheral nerve fibers, some CNS-released neurohormones also regulate tooth and periodontal homeostasis via body fluid circulation. The related studies mainly focus on melatonin, serotonin, and oxytocin (Figure 2).

Melatonin is a tryptophan derivative that regulates circadian rhythm and sleep and has more diverse physiological functions. In experimental periodontitis, melatonin alleviates rats’ periodontal destruction with its anti-oxidative and anti-inflammatory properties [131]. In pulpitis, at physiological concentration, melatonin promotes osteogenic differentiation of dental pulp cells (DPCs) but inhibits their proliferation. It also promotes the apoptosis of DPCs [132,133]. In addition, melatonin, mediated by the MT1 receptor but not MT2, antagonizes the inflammatory response of dental pulp fibroblasts to LPS infection [134]. Nevertheless, it appears melatonin may be helpful for regeneration, but that depends on the disease context and concentration. Its overall role in pulpitis is particularly tricky.

The precursor of melatonin, serotonin, has been the therapeutic target for treating many psychiatric diseases. There are rampant caries rate increases in patients using serotonin reuptake inhibitors (SRIs), a class of anti-depressant. Consistently, the abolition of serotonin signaling induces deformed enamel structure in mice [65]. Serotonin is also shown to facilitate CGRP release from TG neurons in human dental pulp upon stimuli such as tooth extraction, thus participating in the modulation of pain and neurogenic inflammation. This effect is specific to women [135]. Mechanistically, estrogen receptor α (ERα) is co-expressed with serotonin receptor 5HT2A on TRPV1-expressing trigeminal nociceptors. ERα activation in female rats enhances serotonergic potentiation of CGRP release, while its homologous partner, ERβ, does not affect CGRP release [136]. However, this effect differs depending on different estrogen concentrations. A low but steady level of estrogen during diestrus protects against orofacial pain, while an acutely increasing level mimicking estrus potentiates it [136]. In summary, serotonin may protect and even regenerate teeth either directly or indirectly via CGRP, but the latter is sex-specific and affected by individual hormonal states. In periodontitis, serotonin probably enhances periodontal destruction, based on a retrospective study about SRI treatment [137].

Oxytocin is another neurohormone with a peripheral effect on bone remodeling. Although study on oxytocin against a dental background is scarce, unsurprisingly, the provision of oxytocin supplementation to menopausal rats can promote their alveolar bone healing, consistent with its effect on other bone tissue [138]. Oxytocin application also alleviates rats’ periodontitis not only by inhibiting bone resorption but also with its anti-inflammatory and anti-oxidative property, as shown by reduced leukocytes in the gingiva [139]. Oxytocin is also shown to have a peripheral analgesic effect by binding to the oxytocin receptors (OXTRs) on some CGRP^+^ peptidergic nociceptors, decreasing their CGRP release. Emerging evidence suggests this effect is related to its direct binding with TRPV1, a gatekeeper of CGRP release, but whether this binding is inactivating in nature remains elusive [140]. Nevertheless, these clues together suggest oxytocin might become a multi-functional medication for dental and periodontal regeneration, being both analgesic and bone-protective.

### 6.8. Related Applications in Tooth Regeneration

Exogenous supplementation of the neuropeptides or neurohormones regulating tooth homeostasis may promote tooth regeneration or help treat other conditions. CGRP is shown to significantly promote the formation of tertiary dentin in pulpotomized ferret canines, meaning its in vivo function can be replicated by exogenous application [141]. Topical application of CGRP, or mixing it with a biomaterial, may be viable for regenerating dental tissue. As CGRP also promotes the regeneration of peripheral nerve fibers, some scholars hypothesize that local application of CGRP can improve the recovery of proprioception of implants, as the poor osseoperception of dental implants is due to the poor regeneration of nerves around the implant. If it works, it should outperform cellular therapies in terms of safety and reliability, but studies verifying this are still absent [142]. Melatonin prescription could enhance the outcome of non-surgical periodontal treatment in patients with severe periodontitis, owing to its immunomodulatory and antioxidative properties [143]. When combined with an injectable hydrogel scaffold containing beta-tricalcium phosphate, it accelerated the formation and maturation of new bone in critical-sized furcation involvement [144]. Other molecules discussed above may also be beneficial, but related studies are scarce.

In addition, some neuropeptides and neurohormones have not been investigated in the context of dental diseases but have already been proven to be helpful in tooth regeneration. Dopamine as an antioxidant in tissue engineering materials can effectively promote periodontal tissue healing and regeneration [145,146]. Dopamine is released from the CNS and affects various cell behaviors such as adhesion, migration, and survival. A porous poly-aspartamide-based hydrogel loaded with dopamine at a suitable concentration promotes the proliferation of both PDLSCs and SH-SY5Y on the hydrogel, meaning it might help regenerate the defected periodontium with innervation. These data suggest dopamine-loaded poly-aspartamide-based hydrogel holds therapeutic potential in periodontal regeneration [147]. Melanocortin (α-MSH) is an anti-inflammatory neurohormone released by the hypothalamus. It can be covalently coupled to poly-l-glutamic acid (PGA-alpha-MSH) and still be able to reduce inflammatory cytokine expression by LPS-stimulated pulp fibroblasts and promote their adhesion and proliferation, which are favorable for pulp regeneration. A multilayered polyelectrolyte film loaded with PGA-alpha-MSH shows similar effects, making it possible to apply PGA-alpha-MSH to endodontically treated teeth [148]. However, whether these materials actually work in damaged teeth remains to be examined. It is also unknown whether they play a role in physiological tooth reparation because of the absence of basic studies, meaning the tissue they regenerate may not be similar to the physiological state. In summary, applications of this kind require more scrutiny to affirm their efficacy.

## 7. The Nervous System Regulates Dental Stem Cells

Stem cells take a key role in current tissue-engineering methods owing to their multipotency that gives rise to a variety of tissue cells. The question of how the therapeutic potential of dental stem cells can be exploited to the fullest extent in tooth and periodontium regeneration is worth discussing. Here we describe the regulations of the nervous system on dental stem cells in detail (Table 3).

### 7.1. Neural Regulations and Incisor Stem Cells

The incisors of rodents continuously grow throughout their whole lives, which is attributed to the maintenance of epithelial stem cells (ESCs) and mesenchymal stem cells (MSCs), and thus rodent incisors are widely recognized as a useful model for studying stem cells (Table 3).

It has been revealed that the continuous turnover of MSCs in mouse incisors is supported by a neurovascular bundle (NVB) niche, and the innervation is potentially associated with incisor homeostasis and injury repair via MSCs [149]. IANr leads to a significant reduction in incisor eruption rate [13,149], probably due to the decreased MSC count and proliferation [13,149]. The interruption of Wnt signaling resulting from denervation most likely causes a reduction in MSC proliferation. Meanwhile, microarray analysis shows that denervation hinders various signaling pathways, particularly Shh signals, which are secreted from the sensory nerves and modulate the expression of Gli1 in the stem cells near the cervical loop region. Finally, the odontogenic commitment and hard tissue formation are defective, leading to the destruction of incisor homeostasis [149].

In addition, denervation can increase the number of glial cells [149], and abundant Schwann-cell-derived cells located at the incisor damage area contribute to matrix mineralization [150]. This indicates that the innervation regulates incisor MSCs not only through Shh signaling but also via other components. Schwann cell precursors (SCPs) and Schwann cells (SCs) are the major types of glial cells in the peripheral nerves. Lineage tracing reveals incisor SCPs and SCs directly give rise to dental MSCs producing odontoblasts and pulp cells during the process of the development, growth, and regeneration of mouse incisors [150]. Worth mentioning is the glia-derived odontoblasts’ independence of pericytes [150], despite other studies reporting the pericytes as odontoblast-producing MSCs in injury in vivo [163]. This suggests multiple origins for MSCs in rodent incisors and potentially human teeth.

IANr also disrupts incisor ESCs possibly through mesenchymal–epithelial interaction, but to a far minor extent [13]. Nevertheless, ESC niches and differentiation are still directly regulated by Ptch1/2-mediated Shh signaling derived from non-neural sources including transient amplifying cells (TACs) and differentiating ameloblasts [164,165].

The sympathetic nerves appear to have a minor impact on tooth homeostasis. They do not provide Shh for the MSC niches surrounding NVB [149]. Superior cervical ganglionectomy (SCGx) can only cause a slight reduction in dentin thickness [12]. However, it is an important regulator for bone homeostasis. Sympathetic innervation rectifies rat incisor eruption rate and accelerates it when the incisor is injured to avoid occlusion [166].

Human teeth differ biologically from rodent incisors. However, Shh signaling is also shown to regulate the differentiation of human DPSCs. Activation of molar sensory nerves due to injury-induced inflammation causes Shh and CGRP release to elicit these effects, which ultimately promotes reparative dentin formation [125]. Therefore, the Shh signaling pathway is a potential therapeutic target for tooth regeneration involving MSCs.

### 7.2. PNS-Derived Neuropeptides on Dental Stem Cells

DPSCs and periodontal ligament stem cells (PDLSCs) are regarded as important players in dental and periodontal homeostasis owing to their self-renewal and multipotency in tissue healing and regeneration and their regulation of inflammation. Local PNSs, particularly sensory fibers, exert their regulation on dental stem cells via various PNS-derived neuropeptides mentioned before (Table 3).

CGRP is reported to inhibit DPSC proliferation and consequently reduce DPSC quantity but has no noticeable effect on DPSC differentiation [167]. Contradictorily, in another study, CGRP promoted molar pulp cell expansion but inhibited odontoblastic differentiation [125]. Further studies are needed to reconcile these contradicting clues. Nevertheless, CGRP and Shh together orchestrate DPSCs’ response to injury in vitro, as Shh promotes odontoblastic differentiation which offsets CGRP’s inhibitory effect, while CGRP promotes proliferation [125]. Therefore, if CGRP is to be applied in tooth regeneration, combined use with Shh or other molecules should be considered, especially when coupled with DPSCs. Reversely, proinflammatory cytokines and chemokines from DPSCs sensitize trigeminal neurons by binding to their CXCR2 receptor and agonizing CGRP release in human pulp explants [168]. This is consistent with the previous observation that DPSC proliferation and immunomodulation are altered in inflammatory environments, meaning caution should be taken when treating dental tissue defects caused by inflammatory diseases with biomaterials involving DPSCs.

SP, a potent proinflammatory neuropeptide, modulates PDLSCs directly and indirectly. It is reported that PDLSCs incubated with SP produce CCL20/MIP-3α by triggering p38 and ERK/MAPK signaling. CCL20 and its receptor CCR6 show marked upregulation in diseased periodontal tissue and promote the selective recruitment of T cells [157]. In bone remodeling, SP is usually regarded as an osteoclastogenic inducer via regulating PDLSCs. However, it is also reported to protect osteogenesis. At a concentration similar to that in GCF in periodontitis patients, SP activates HO-1 and Nrf-2, subsequently upregulates the osteogenic differentiation of human PDLSCs [158]. Therefore, further studies are still needed to clarify its global role in periodontitis progression. Nevertheless, these data demonstrate that modulation of SP may be important for periodontal regenerative therapy with MSCs.

Components from sympathetic innervation also regulate PDLSC differentiation. Treatment of PDLSCs with NPY but not SP has a moderate osteogenic effect. In reverse, PDLSCs stimulated by IL-1b and TNF-α upregulated the NPY Y1 receptor and downregulated the SP receptor [117], which implicates a protective response in periodontal inflammation. Therefore, pretreatment of PDLSCs should be considered when regenerating inflammation-compromised periodontium with them.

### 7.3. The Role of Peripheral Non-Neuron Cells in Dental Stem Cells

In addition to the neuropeptides from neurons, non-neuron cells also play important roles in the modulation of dental stem cells, also predominantly via their secreta (Table 3).

In humans, SCs regulate pulpal homeostasis by releasing extracellular vesicles (EVs). SC-EVs promote DPSCs’ multipotency by transferring transcription factor Oct4, which subsequently upregulates Sox2 and Nanog. These EVs are also enriched with TGFβ, which helps with DPSCs proliferation by activating the TGF-β/Smad and TGF-β/MAPK signaling [151]. Aside from regulating local DPSCs, SC-EVs are capable of recruiting endogenous stem cells to facilitate pulp regeneration. They can promote the migration of DPSCs, neurons, and vascular endothelioid cells; accelerate neurite outgrowth and vessel formation; and enhance the osteogenic differentiation of DPSCs in vitro. These effects are mainly achieved by transporting SDF-1 (CXCL12) to activate CXCR4 on target cells. In vivo study reveals SC-EVs and SDF-1 can effectively promote the formation of pulp–dentin complex-like structures [152].

SCs also regulate periodontal wound healing. The migration of SCs to injury sites is promoted by PDLSC-expressed GDNF, and SCs subsequently interact with local PDLSCs to enhance their osteogenic differentiation probably via the ERK1/2 signaling pathway [159]. This reveals that SCs regulate periodontal healing by reciprocally interacting with PDLSCs at injury sites. This feedback loop between SCs and dental stem cells is also implicated elsewhere. Gingiva-derived mesenchymal stem cell (GMSC)-derived EVs induce the repair phenotype of SCs in vitro, which in turn facilitates regeneration and functional recovery of injured peripheral nerves. c-JUN is a key transcription factor driving the activation of the repair phenotype of SCs in this process [169].

Above all, SCs are potent regulators of the homeostasis and regeneration of the teeth and periodontium. Combined application of dental MSCs and SCs, or pretreatment with SCs derivatives, will potentially benefit the regeneration outcome of biomaterials using MSCs, especially in improving the re-innervation of the bio-engineered teeth or periodontium.

### 7.4. CNS-Released Neurohormones and Stem Cells

As described in Section 6, CNS-released neurohormones regulate tooth and periodontal homeostasis, which can also be achieved by affecting stem cells (Table 3).

The anti-inflammatory property of melatonin appears consistent in the dental context, as melatonin treatment increases TGF-β production by DPSCs, which subsequently impedes T-cell proliferation upon mitogen stimuli [154]. The other melatonin actions on DPSCs’ biological properties seem to be concentration-dependent. It is reported an intermediate concentration of melatonin promotes the proliferation and migration of DPSCs while failing to induce their osteogenic differentiation [154]. However, other recent studies reveal that at intermediate concentrations, melatonin promotes osteogenic differentiation of DPSCs by suppressing DNA methylation [155], as well as via COX-2/NF-κB and MAPK signaling pathways. It also increases DPSCs proliferation [153]. At lower concentrations, melatonin inhibits DPSCs proliferation and induces DPSCs neuronal differentiation by the activation of the Hippo pathway [156]. The precursor of melatonin, serotonin, is mainly stored in platelets, whose co-release of serotonin and dopamine upon injury stimulation promotes reparative dentin formation via DPSCs [170]. These pieces of evidence demonstrate the protective and pro-regenerative function of melatonin and serotonin in dental pulp. They may be applicable in the regeneration of pulp and bone or nerve repair with DPSCs, but concentration should be particularly considered when applying melatonin.

As for PDLSCs, melatonin suppresses their osteogenic differentiation at physiological concentration by altering mitochondrial dynamics manifested as increased ROS and mitochondrial fission [161]. However, this action is reversed in bone mesenchymal stem cells (BMSCs), where osteogenic differentiation is rescued by melatonin at pharmacological concentration by activating AMPK and upregulating FOXO3a and RUNX2 [162]. Nevertheless, melatonin still holds pharmacological utility in promoting osteogenesis and antioxidation. In addition, melatonin is also shown to effectively rejuvenate long-term ex vivo expanded PDLSCs by restoring their autophagy via PI3K/AKT/mTOR signaling [160]. Together, these data reveal the application potential of melatonin in dental stem cell-based regenerative therapy.

### 7.5. Related Application in Tooth Regeneration

Scaffold material incorporated with dental stem cells is a promising direction. PDLSCs and SH-SY5Y neuroblastoma cells can migrate into the depth of a porous poly-aspartamide-based hydrogel loaded with dopamine and degrade it. The hydrogel is designed for periodontal regeneration; therefore, if it is applied together with these cells, better regeneration capacity may be attainable [147]. In addition, the odontogenic differentiation of DPSCs is effectively promoted by a mixture of melatonin and equine bone blocks, meaning combined use of MSCs with biomaterial scaffolds and neurohormones not only adds their regenerative capacity but even augments it, which will be favorable for regeneration therapy [171]. However, there is prolonged debate on the safety of stem cell-based therapy. In the context of tooth regeneration, related concerns such as infection and tumorigenesis still exist despite the application being only topical.

We also notice that there is currently no study on the neural regulation of stem cells from apical papilla (SCAPs) and dental follicle stem cells (DFSCs). This is probably because they are harder to obtain than DPSCs and PDLSCs since DFSCs exist in the DF of tooth germ and SCAPs in the apical papilla, the dental mesenchyme underneath developing tooth root responsible for root elongation. SCAPs and DPSCs both originate from the dental papilla, and DFSCs are direct precursor cells for periodontal tissues [172]. This means neural regulations on DPSCs and SCAPs and those on PDLSCs and DFSCs may be somewhat alike. SCAPs together with PDLSCs regenerated functional root–periodontal complex [173], and DFSCs can regenerate periodontium in vivo [174]. Therefore, they may be very promising candidates for tooth and periodontium regeneration. Studies on the neural regulations on them will be meaningful for improving hard tissue and re-innervation outcome of tooth and periodontium regeneration.

## 8. Conclusions

The tooth–periodontium complex has developmental homology with the nervous system. Evidence abounds revealing the participation of nerves in tooth–periodontium development and homeostasis. Some molecules mediating these processes are applied in tooth regeneration studies and have proven their efficacy. This review summarizes recent studies on the above aspects and sheds light on many blank spaces to be filled by future studies.

Research gaps include the timing and distribution of sympathetic innervation towards the developing tooth germ, whether sensory innervation is indispensable in pre-mineralization tooth development, and the neural regulation mechanism in tooth eruption. Currently, one major strategy for tooth regeneration is to mimic the natural development of the tooth using tooth germ cells, and thus understanding tooth developmental mechanisms will help develop related regenerative therapy. It will also instruct the application of molecules implicated in tooth development to help better regenerate teeth.

In tooth and periodontal diseases, the roles of many neuropeptides and neurohormones are controversial and may depend on experimental settings. The roles of many other neurohormones and neuropeptides, such as α-MSH and dopamine, remain to be examined in dental contexts. As tissue engineering is another current major direction for tooth regeneration, a better understanding of these molecules will help select more candidates to be used in biomaterials. Even if some existing tooth regeneration studies applied neurogenic molecules, their feasibility, safety, efficacy, and potential side effects need to be verified in clinical studies.

These unknowns hamper the translation of physiological mechanisms into application in regenerative dentistry. This review, by summarizing current research advances and pointing out research gaps in this field, will be instructive for future research directions and will provide more clues for therapeutic innovation.

## Figures and Tables

**Figure 1 ijms-23-14150-f001:**
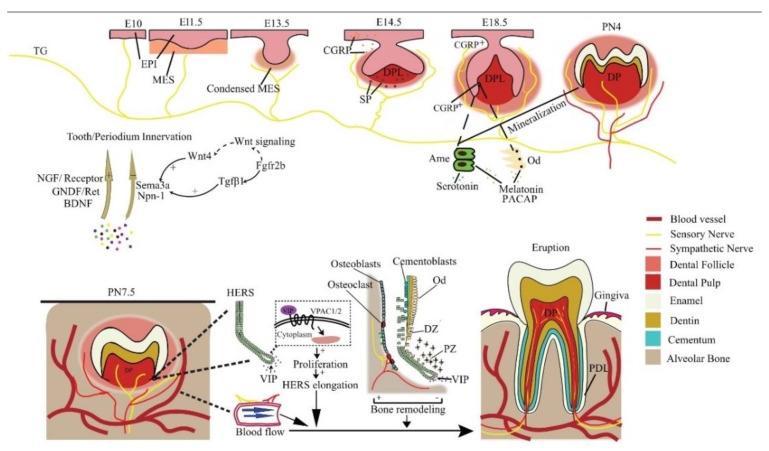
Reciprocal regulation between developing tooth and its innervation. NGF, GNDF, and BDNF promote nerve growth, while Sema3a and Npn-1 suppress it and ensure normal patterning. In a mouse embryo, the pioneering trigeminal ganglion (TG) fibers first arrive at the maxillary oral epithelium on E10. On E14.5, SP is detected and is essential for tooth development to continue. The sensory fibers grow into the dental follicle on E18.5. CGRP, PACAP, melatonin, and serotonin regulate tooth mineralization. Sensory fibers enter the dental pulp on PN4, while the sympathetic fibers enter it on PN 9, after root formation has begun. Neural VIP promotes the elongation of HERS to stimulate the differentiation of DF cells, resulting in root growth. Periodontal nerves regulate bone remodeling as well as the local blood flow in alveolar sockets to affect tooth eruption. EPI = epithelium; MES = mesenchyme; Ame = ameloblast; Od = odontoblast; NGF = nerve growth factor; DPL = dental papilla; DP = dental pulp; PDL = periodontal ligament; CGRP+ = CGRP positive cells; DZ = differentiation zone; PZ = proliferation zone.

**Figure 2 ijms-23-14150-f002:**
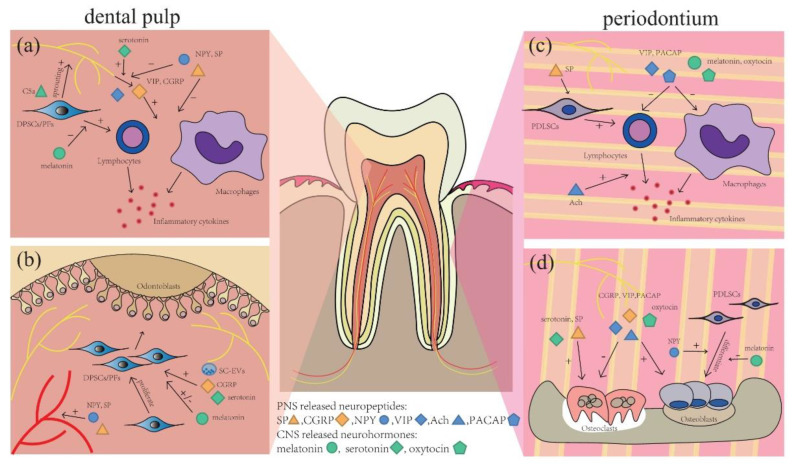
Neural regulation in tooth and periodontal diseases. During pulpitis, (**a**) C5a stimulates neurotrophins release from DPSCs/pulp fibroblasts (PFs) to induce nerve fiber sprouting. Melatonin, VIP, and CGRP are anti-inflammatory, while NPY and SP are proinflammatory. Serotonin enhances CGRP release, while NPY attenuates it. (**b**) Serotonin, melatonin, CGRP, and extracellular vesicles from Schwann cells (SC-EVs) can regulate the formation of restorative dentin by affecting DPSCs/PFs proliferation and differentiation. SP and NPY contribute to pulp hyperemia. In the periodontium, (**c**) PACAP, VIP, melatonin, and oxytocin are anti-inflammatory, while Ach and SP conduce to periodontal inflammation. (**d**) CGRP, PACAP, VIP, and oxytocin directly promote bone regeneration, and NPY promotes osteogenesis by PDLSCs. Meanwhile, SP, serotonin and melatonin induce periodontal destruction.

**Table 1 ijms-23-14150-t001:** Nerves in the pulp.

Nerve Fibers	Regional Distribution	Function
sensory	Raschkow plexus Throughout the pulp In the dentinal tubule	Highly specialized nociceptors
sympathetic	SCG→TG→sensory fibers→pulp SCG→inferior/superior alveolar artery→pulp Around pulp arterioles Raschkow plexus	Regulate pulp blood flow

**Table 2 ijms-23-14150-t002:** Nerves in the periodontal ligament.

Nerve Fibers	Regional Distribution	Function
sensory	Throughout the PDL As bundles around blood vessels near the alveolar bone As free endings near the cementum	Nociceptors and mechanoreceptors Regulate blood flow
autonomic	Few	Regulate blood flow

**Table 3 ijms-23-14150-t003:** The stem cells related to the tooth–periodontium complex are regulated by the nervous system.

Stem cells	Neural Regulations	Biological Effects	Molecular Mechanism	Ref.
MSCs (Incisor)	IAN secretion	Promoting MSC proliferation	Wnt signaling	[149]
		Sustaining Gli1 expression and odontogenic commitment of MSCs	Shh signaling	[149]
	SCs/SCPs	Giving rise to MSCs that generate odontoblasts and pulp cells	SCs and SCPs differentiation	[150]
ESCs (Incisor)	IAN	Regulating ESC niches and differentiation	Mesenchymal–epithelial interaction	[13]
DPSCs	Shh & CGRP	Shh promotes odontoblastic differentiation and CGRP promotes proliferation of DPSCs	CGRP/Shh signaling	[125]
	SC-EVs	Promoting DPSC multipotency	Oct4/Sox2/Nanog	[151]
		Promoting DPSC proliferation	TGF-β/Smad signaling TGF-β/MAPK signaling	[151]
		Promoting DPSC migration and osteogenic differentiation	SDF-1/CXCR4	[152]
	Melatonin (intermediate concentration)	Promoting DPSC proliferation and osteogenic differentiation	COX-2/NF-κB signaling p38/ERK MAPK signaling	[153]
		Promoting DPSC migration and proliferation	-	[154]
		Increasing TGF-b production by DPSCs to suppress T-cell proliferation upon stimuli	-	[154]
		Promoting DPSC osteogenic differentiation	Suppressing DNA methylation	[155]
	Melatonin (low concentration)	Inhibiting DPSC proliferation and inducing DPSC neuronal differentiation	Hippo pathway	[156]
PDLSCs	SP	PDLSCs produce CCL20/MIP-3α to recruit T cells	p38 and ERK/MAPK signaling	[157]
	SP	Promoting PDLSC osteogenic differentiation	SP/HO-1/Nrf-2	[158]
	NPY	Moderate osteogenic effects	-	[117]
	SCs	Promoting osteogenic differentiation of PDLSCs	ERK1/2 signaling	[159]
	Melatonin (low concentration)	Rejuvenating long-term ex vivo expanded PDLSCs by restoring their autophagy	PI3K/AKT/mTOR signaling	[160]
		Suppressing PDLSC osteogenic differentiation	Increasing mitochondrial fission	[161]
BMSCs	Melatonin (intermediate concentration)	Promoting BMSCs osteogenic differentiation	AMPK/FOXO3a/RUNX2 signaling	[162]

## Data Availability

Not applicable.

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
