# Peer review of "Neural Regulations in Tooth Development and Tooth–Periodontium Complex Homeostasis: A Literature Review"

_ijms, 2022, doi:10.3390/ijms232214150_

Round 1
Reviewer 1 Report
The article entitled "Neural regulations of tooth development and tooth-periodontium homeostasis: a review" is very interesting to me about the contents, and I think it is valid for publication. However, the topics covered are various, and even if divided into paragraphs, they are not always easy to read. A check of these points is suggested, and if possible a reorganization of the topics dealt with.
Author Response
November 10, 2022
ijms-1987711
Response to Reviewers’ Comments:
Reviewer1 The article entitled "Neural regulations of tooth development and tooth-periodontium homeostasis: a review" is very interesting to me about the contents, and I think it is valid for publication. However, the topics covered are various, and even if divided into paragraphs, they are not always easy to read. A check of these points is suggested, and if possible a reorganization of the topics dealt with.
Response: Thank you for your kind advice. We have addressed the structure problem of our original manuscript by making the following adjustments in the revised manuscript:
- In the introduction, we define the purpose of this article is to provide more clues and inspiration for improving the outcome of tooth and periodontal tissue regeneration, whether it is based on mimicking developmental process or physiological reparation. Provided this, we believe the topics of our revised article will appear more logically connected and easier to comprehend.
- We articulated the purpose of this review at the end of the abstract, in the last two paragraph of the introduction, and in the conclusion section.
- We added analyses on how these basic researches on neural regulation mechanisms may be related to tooth and periodontal regeneration and their potential concerns. These analyses are presented in orange font in the revised manuscript.
- We disassembled the major section 8 “Regeneration strategies: inspiration from neural regulations” in the original manuscript, into subtitles of different major sections in the revised version (subtitle 3.2.3, 4.3, 5.3, 6.8, 7.5), basing on the different biological mechanisms of the regeneration studies we’re referring to. We believe it will highlight the linkage between the regeneration studies and the basic researches in those major sections, thus to reinforce the purpose of our review.
- We renamed the different sections of this review to better guide the readers through our plot.
Reviewer 2 Report
The topic of the paper is very interesting. The study of literature is very extensive and required a lot of work. However, there are some problems that I would like to point out.
The type of review should be mentioned in the title. MDPI title and abstract format should be adopted.
The abstract is not well organized. The addressed scientific question is not mentioned. The statement: “This review summarizes these findings in order to provide therapeutic strategies for dental diseases and tooth regeneration.”, can mislead the reader, because the paper is way far from treatment strategies. The expression used at the end of the introduction “will provide more clues for the regeneration of tooth and the therapy of dental disease”, it is much closer to what it offers.
The main and secondary aims of this review are not clearly stated at the end of the introduction. There is no Conclusion section to suport the objectives.
Caption for Figure 1 and Figure 2, too large. The explanations should be placed in the main text.
The affirmation: “But to date, no study has focused on the mechanism for serotonin stimulating tooth development”, Line 265-266 it is not well documented. After a brief search in the literature, there are plenty of studies published.
Example:
J R Moiseiwitsch , J M Lauder. Stimulation of murine tooth development in organotypic culture by the neurotransmitter serotonin. Arch Oral Biol. 1996 Feb;41(2):161-5. doi: 10.1016/0003-9969(95)00117-4.
J R Moiseiwitsch. The role of serotonin and neurotransmitters during craniofacial development
Please check the paper for typing and/or content errors like in line 527: “Melatonin has a direct action on osteoblast and osteoblast”, line 550: “Erαactivation in female rats enhances serotonergic potentiation of CGRP release”.
Citation is missing line 634: “CGRP is reported to inhibit DPSCs proliferation and consequently reduce their quantity but have no noticeable effect on their differentiation [135]. Contradictorily, in another study, CGRP promoted molar pulp cells expansion but inhibited odontoblastic differentiation.” [no reference for the second study, that was mentioned]
The results are poorly presented and confusing, I recommend a clearer presentation, diagrams, tables. Many details related to embryological and histological aspects and very few clinical and practical implications brought into discussion.
The article is mostly written like a book chapter. Hard to read and follow. At the end, the authors have a section entitled summary instead of writing a section of conclusions. This confirms the book chapter structure and the minimal contribution of the authors in terms of scientific progress.
Author Response
November 10, 2022
ijms-1987711
Response to Reviewers’ Comments:
Reviewer 2
The topic of the paper is very interesting. The study of literature is very extensive and required a lot of work. However, there are some problems that I would like to point out.
1.The type of review should be mentioned in the title. MDPI title and abstract format should be adopted.
Response: We apologize for the format errors in the original manuscript. We have made the type of our review clear in the title, and have adopted the suggested format of title and abstract in the revised manuscript.
2.The abstract is not well organized. The addressed scientific question is not mentioned. The statement: “This review summarizes these findings in order to provide therapeutic strategies for dental diseases and tooth regeneration.”, can mislead the reader, because the paper is way far from treatment strategies. The expression used at the end of the introduction “will provide more clues for the regeneration of tooth and the therapy of dental disease”, it is much closer to what it offers.
Response: We’re sorry that the purpose of our review is not well explained in the abstract and may lead to misunderstanding in the original manuscript. In the revised manuscript, we have restructured the abstract to make our purpose clear. The purpose of this article is to provide more clues and inspiration for improving the outcome of tooth and periodontal tissue regeneration, whether it is based on mimicking developmental process or physiological reparation.
3.The main and secondary aims of this review are not clearly stated at the end of the introduction. There is no Conclusion section to suport the objectives.
Response: Thank you for your suggestion, we have changed the content and structure of the introduction in the revised manuscript, providing background information about tooth regeneration and clearly stated the purpose of this review in the last two paragraph of the introduction. Our main aim is to provide more clues basing on the existing basic researches to improve the outcome of tooth and periodontal regeneration, and the secondary aim is to point out gaps in current basic researches to be filled in future. We have also added a conclusion section instead of summary to support our objectives.
4.Caption for Figure 1 and Figure 2, too large. The explanations should be placed in the main text.
Response: We have cut the caption for Figure 1 and Figure 2 to be shorter in the revised manuscript. We’re sorry it may not be appropriate to place the explanations of the figures into the main text, because they’re already a summary of the main text contents.
5.The affirmation: “But to date, no study has focused on the mechanism for serotonin stimulating tooth development”, Line 265-266 it is not well documented. After a brief search in the literature, there are plenty of studies published.
Example:
J R Moiseiwitsch , J M Lauder. Stimulation of murine tooth development in organotypic culture by the neurotransmitter serotonin. Arch Oral Biol. 1996 Feb;41(2):161-5. doi: 10.1016/0003-9969(95)00117-4.
J R Moiseiwitsch. The role of serotonin and neurotransmitters during craniofacial development
Response: We apologize for this false affirmation. After re-conducting the research in the literature, we have added related content on serotonin promoting tooth development in the revised manuscript, in line 279-282. The related literatures have been cited in this review ([62], [63]). Thanks for your pointing!
6.Please check the paper for typing and/or content errors like in line 527: “Melatonin has a direct action on osteoblast and osteoblast”, line 550: “ enhances serotonergic potentiation of CGRP release”.
Response: Thank you for pointing out these errors. We have thoroughly checked the typing and content errors of the article, and corrected all errors of this kind in the revised manuscript, such as the “Erαactivation in female rats” in line 564. As for the other error that you pointed out in line 527 in the original manuscript, related content has been removed in the revised version after reconsideration to better fit our purpose to inspire regeneration study.
7.Citation is missing line 634: “CGRP is reported to inhibit DPSCs proliferation and consequently reduce their quantity but have no noticeable effect on their differentiation [135]. Contradictorily, in another study, CGRP promoted molar pulp cells expansion but inhibited odontoblastic differentiation.” [no reference for the second study, that was mentioned]
Response: Thank you for pointing out these errors. We have thoroughly checked the citation errors of the article, and corrected all errors of this kind in the revised manuscript, including the one you pointed out in line 679.
8.The results are poorly presented and confusing, I recommend a clearer presentation, diagrams, tables. Many details related to embryological and histological aspects and very few clinical and practical implications brought into discussion.
Response: Thank you for your suggestions. In the revised manuscript, we added several tables (Table 1- Table 3) to better present the research results. We discussed the relationship between the nervous systems and dental diseases, including caries, pulpitis, occlusal trauma, gingival inflammation and so on, in parts 5 and 6. It is helpful for understanding clinical signs and symptoms. The inspiration and clues for tooth regeneration have also been mentioned here. In fact, most of the current regeneration studies are only proof of concepts and are still way far from clinical application. Therefore, unfortunately, we find it hard to bring in too much clinical and practical aspects. We are very sorry for that.
9.The article is mostly written like a book chapter. Hard to read and follow. At the end, the authors have a section entitled summary instead of writing a section of conclusions. This confirms the book chapter structure and the minimal contribution of the authors in terms of scientific progress.
Response: We apologize for the text book-like structure of our original manuscript. In our revised manuscript, we wrote a conclusion section (section 8) instead of the summary, to support the purpose and highlight the scientific value of our review.
Round 2
Reviewer 2 Report
Thanks to the authors for the effort performing the changes.